# Local Immune Biomarker Expression Depending on the Uterine Microbiota in Patients with Idiopathic Infertility

**DOI:** 10.3390/ijms24087572

**Published:** 2023-04-20

**Authors:** Natalya I. Tapilskaya, Alevtina M. Savicheva, Kira V. Shalepo, Olga V. Budilovskaya, Aleksandr M. Gzgzyan, Olesya N. Bespalova, Tatiana A. Khusnutdinova, Anna A. Krysanova, Kseniia V. Obedkova, Galina Kh. Safarian

**Affiliations:** D.O. Ott Research Institute of Obstetrics, Gynecology and Reproductive Medicine, 199034 St. Petersburg, Russia; tapnatalia@mail.ru (N.I.T.); savitcheva@mail.ru (A.M.S.); 2474151@mail.ru (K.V.S.); o.budilovskaya@gmail.com (O.V.B.); agzgzyan@gmail.com (A.M.G.); shiggerra@mail.ru (O.N.B.); husnutdinovat@yandex.ru (T.A.K.); krusanova.anna@mail.ru (A.A.K.); obedkova_ks@mail.ru (K.V.O.)

**Keywords:** defensins, cytokines, transforming growth factor, infertility, microbiota, endometrium

## Abstract

The endometrium has traditionally been considered sterile. Nowadays, active studies are performed on the female upper genital tract microbiota. Bacteria and/or viruses colonizing the endometrium are known to alter its functional properties, including receptivity and embryo implantation. Uterine cavity inflammation caused by microorganisms leads to disrupted cytokine expression, which, in turn, is mandatory for the successful implantation of the embryo. The present study assessed the vaginal and endometrial microbiota composition and its relation to the levels of cytokines produced by the endometrium in reproductive-aged women complaining of secondary infertility of unknown origin. The multiplex real-time PCR assay was applied for vaginal and endometrial microbiota analysis. The quantitative measurement of endometrial α-defensin (DEFa1), transforming growth factor (TGFβ1), and basic fibroblast growth factor (bFGF2) was carried out using the ELISA (Cloud-Clone Corporation (Katy, TX, USA; manufactured in Wuhan, China). A reliable decline in endometrial TGFβ1 and bFGF2 and an increase in DEFa1 were demonstrated in women with idiopathic infertility when compared to fertile patients. However, TGFβ1, bFGF2, and DEFa1 expression correlated reliably only with the presence of *Peptostreptococcus* spp. and HPV in the uterine cavity. The obtained results highlight the importance of local immune biomarker determination in the assessment of certain bacteria and viruses’ significance as causative agents of infertility.

## 1. Introduction

At present, there is evidence indicating the presence of endometrial microbiota [1]. The degree of uterine contamination depends on the qualitative and quantitative composition of microorganisms, as well as on the duration of colonization, determining the pattern of consecutive and possibly reversible events in endometrial function, including receptivity, blastocyst nesting, and desquamation during menstrual and/or menstrual-like bleeding [2]. However, which particular endometrial microorganisms and/or microbial associations exert the infectious agent’s role and their contribution in endometrial pathology remains unclear [3]. To date, scientific data has been accumulated on the potential negative impact of persistent endometrial microorganisms as a marker of chronic endometritis, the cause of recurrent implantation failure and spontaneous miscarriage [4].

In the review by Sirota et al. (2014), uterine cavity inflammation caused by the presence of bacteria was reported to alter the cytokine pathways necessary for successful blastocyst development and implantation [5]. An inflammatory cytokine reaction in endometriosis is known to significantly affect the reproductive outcome [6]. In addition, some pro-inflammatory cytokines, such as IL-6, and anti-inflammatory cytokines have been reported to be associated with various gynecological conditions, i.e., polycystic ovary syndrome (PCOS), tubal factor infertility, or infertility of unknown origin [7]. 

Defensins represent cationic peptides consisting of 18–45 amino acids, active against bacteria, fungi, and a large number of both enveloped and non-enveloped viruses [8]. These molecules play a key role in the body’s first-line protection against infectious agents [9]. More recently, the antiviral activity of some defensins was demonstrated [10]. Thus, the anti-HIV-1 activity of defensins was revealed, leading to growing opportunities in HIV investigation [11]. In addition, patients lacking α-defensins (specific granule deficiency syndrome) have increased susceptibility to bacterial infections [12]. 

The transforming growth factor beta (TGF-beta, TGFβ) represents a secreted extracellular multifunctional cytokine produced by all lines of white blood cells, controls an immense number of cellular responses, and participates in the development and homeostasis of most human body tissues [13]. The transforming growth factor beta gene (TGFB1 gene) maps to human chromosome 19. The regulation of inflammatory processes is believed to be one of the main functions of TGF-beta. In addition, this protein also plays a decisive role in stem cell differentiation, as well as T-cell regulation and differentiation [14]. TGFβ1 represents a polypeptide member of the transforming growth factor beta superfamily of cytokines and controls a number of cellular functions, including cell growth, proliferation, differentiation, and apoptosis. All the above-mentioned has a detrimental impact on the reproductive function of mammalians [15]. 

Fibroblast growth factor (FGF) is a representative growth factor that plays a crucial role in angiogenesis and early embryonic development in humans [16]. Until now, there is still no clear understanding of the impact of endometrial microbiota on antimicrobial peptides and cytokine expression. Thus, the present study aimed to assess the composition of vaginal and endometrial microbiota and its association with the levels of antimicrobial peptides and cytokines in endometrium samples of patients with infertility.

## 2. Results

### 2.1. Vaginal and Endometrial Microbiota Composition

In the infertile group of patients (n = 65), the endometrial microbiota analysis revealed a lack of microorganisms in 6 (9.23%) endometrial samples. *Trichomonas vaginalis*, *Neisseria gonorrhoeae*, *Chlamydia trachomatis*, HSV-1/2, CMV, EBV, and HHV-6 were not found in any endometrial samples. 

*Lactobacillus* spp. (41.5%), *Bacteroides* spp., *Porphyromonas* spp., *Prevotella* spp. (38.8%), *Gardnerella vaginalis* (35.4%), *Enterobacteriaceae* spp. (18.5%), as well as *Enterococcus* spp. (16.9%) and *Bifidobacterium* spp. (16.9%) were most frequently identified endometrial microorganisms. 

*Megasphaera* spp., *Velionella* spp., *Peptostreptococcus* spp., *Sneathia* spp./*Leptotrihia* spp./*Fusobacterium* spp., *Enterobacteriaceae* spp., *Enterococcus* spp., *Candida* spp. and *Ureaplasma parvum* were the most commonly verified combination of microorganisms.

Table 1 contains data on microbial concordance in both the endometrium and vagina. The highest microbiome concordance in the endometrium was demonstrated for *Bifidobacterium* spp. (36.8%), *Gardnerella vaginalis* (33.3%), *Mycoplasma hominis* (33.3%), *Lactobacillus* spp. (28.1%), Human papillomavirus (HPV) (26.7%), and *Enterobacteriacea* spp. (23.1%). Discordance in the endometrium microbiota was verified for *Staphylococcus* spp., *Streptococcus agalactiae*, *Candida* spp., *Ureaplasma parvum*, *Trichomonas vaginalis*, *Haemophilus* spp., and *Anaerococcus* spp.

### 2.2. Evaluation of Cytokines and α-Defensin in Endometrial Samples

A reliable decrease in endometrial TGFβ1 and bFGF2 concentrations in combination with increased DEFa1 concentration was demonstrated in the infertile group relative to the fertile group (Table 2).

The intragroup analysis of endometrial cytokines and α-defensin concentrations revealed their association with the history of recurrent herpes simplex virus (HSV) infection, ectopic pregnancy, and/or miscarriage. Table 3 contains data on cytokines and α-defensin concentrations in the endometrium samples of women with idiopathic infertility depending on microorganisms verified.

### 2.3. Patient Follow-Up

Eight (12.3%) cases of spontaneous pregnancy in the infertile group of patients were observed during the follow-up period. The retrospective analysis demonstrated reliably increased endometrial TGFβ1 (7.87 pg/mL [6568; 11.42]) and bFGF2 (36.4 pg/mL [28.3; 42.6]) concentrations. No significant differences were observed in levels of α-defensin. Table 4 represents the retrospectively determined data on endometrial microbiota composition in patients with achieved pregnancy.

## 3. Discussion

The implantation of an embryo for pregnancy initiation represents the most important stage in the onset of pregnancy and involves a process of superimposition of a blastocyst-derived signature onto the receptive endometrium during the mid-secretory phase of the menstrual cycle, the so-called “window of receptivity”. Both a receptive endometrium and timely signaling between the blastocyst and endometrium determine the success of implantation [17]. For instance, such endometrial factors as calcitonin [18], lysophosphatidic acid (LPA) [19], heparin-binding epidermal growth factor (HB-EGF) [20], leukemia-inhibiting factor (LIF) [21], and epidermal growth factor (EGF) [16] promote embryo implantation. At the same time, the endometrium undergoes remodeling in response to a variety of factors secreted by the embryo, i.e., hCG [6], IL-1 [15], BMP2 [18], IGF1 [20], FGF2 [16], and WNT [22]. The immune environment changes [19] and tissue remodeling [21] occurs at the contact site of the embryo with the uterine basal lamina.

Endometrial microbiota is known to play a significant role in pregnancy establishment [19]; however, the specification of the qualitative and quantitative composition of the resident endometrial microorganisms still requires the accumulation of scientific knowledge [2]. In our study, only 9.2% of the endometrium samples did not contain microorganisms, and only 29.2% of the subjects analyzed exerted 1 operational taxonomic unit. Further analysis revealed the most common taxa combination of *Megasphaera* spp./*Velionella* spp. with *Peptostreptococcus* spp. and *Sneathia* spp./*Leptotrichia* spp./*Fusobacterium* spp. Apparently, symbiotic microbial interactions ensure certain taxa combinations.

The highest concordance between the upper (endometrial) and lower (vaginal) microbiota was identified for *Bifidobacterium* spp./*Gardnerella vaginalis* (36.8%) and *Bifidobacterium* spp./*Mycoplasma hominis* (33.3%). In addition, high concordance was observed for *Lactobacillus* spp. (28.1%), *HPV* (26.7%), and *Enterobacteriaceae* (23.1%). The data obtained indicate a mainly ascending mode of upper genital tract seeding, such as migration of bacteria through the cervical canal. Previously, lower clinical pregnancy rates were demonstrated after bacteria were grown from the catheter tip after embryo transfer during the assisted reproductive technology procedure [23]. Swidsinski et al. (2013) demonstrated the presence of endometrial polymicrobial biofilms containing *Gardnerella vaginalis* in a fair number of women with bacterial vaginosis [24].

It should be emphasized that many receptivity biomarkers secreted by the endometrium and responsible for trophoblast invasion exist in two variants and the transition into the active variant is driven by hormonal, cytokine, and integrin signals from the implanting embryo [25,26]. For instance, human chorionic gonadotropin (hCG) is known to regulate the expression of four proteins in the human endometrium, namely, IL-11, CXCL10, GMCSF, and FGF2. It has been demonstrated that the FGF2 expression in the glandular epithelium of the human endometrium in vivo increases during the secretory phase of the menstrual cycle and remains high during the first trimester of pregnancy. FGF2 ensures the adhesion of trophectoderm to the inner cell mass (ICM) and type IV collagen [27]. 

The inducible bacteriolytic proteins (IBPs), represented by three proteins discovered in the giant silk moth Hyalophora cecropia, were first reported by D. Hultmark et al. in 1980 [28]. Five years later, T. Ganz et al. first described IBPs produced by human neutrophils, and the term “human defensin” was proposed [29]. At present, IBPs include human defensins, cathelicidins, and some other chemical groups implementing the antimicrobial potential of neutrophils and epithelial cells [30].

Human defensins are generally distinguished as arginine-, lysine-, histidine-, and cysteine-rich short peptides containing 28–51 amino acid residues. Six cysteine residues form one to three disulfide bonds promoting a three-dimensional molecular structure. The arrangement of disulfide bonds dictates the structural differences between defensins, generally divided into the alpha (α), beta (β), and theta (τ) subfamilies [31]. For instance, α-defensins consist of a triple-stranded β-sheet structure established by three disulfide bonds in positions Cys1-Cys6, Cys2-Cys4, and Cys3-Cys5 [9].

At least six human α-defensins are distinguished: four of them are primarily expressed in bone marrow cells, mainly promyelocytes, while two others are abundant in the small intestine’s Paneth cells. In addition, some amount of α-defensins is produced by epithelial cells providing a role in mucosal immunity. α-defensins are encoded by the HDEFA1 and HDEFA3 genes and their expression in neutrophils is proportional to the number of gene copies. In different populations the number of gene copies may vary, reaching a maximum of 12. α-defensins are known for the wide range of antimicrobial activity against both Gram-positive and Gram-negative bacteria such as *Staphylococcus epidermidis*, *S. aureus*, *MRSA*, *E. coli*, *Salmonella typhimurium*, *Proteus mirabilis*, *P. vulgaris*, *Bacillus subtillis*, *Pseudomonas aeruginosa*, *Listeria monocytogenes*, *Burkholderia cepacia*, *Stenotrophomonas maltophilia*, and *Candida albicans*, as some particular viruses [9].

According to the results of our study, a reliable decrease in endometrial TGFβ1 and bFGF2 and an increase in DEFa1 were demonstrated in women with idiopathic infertility when compared to fertile patients. However, TGFβ1, bFGF2, and DEFa1 expression correlated reliably only with the presence of *Peptostreptococcus* spp. and HPV in the endometrium. Additionally, the analysis of endometrial cytokines and α-defensin concentrations revealed their reliable correlation with the history of recurrent HSV infection, ectopic pregnancy, and/or miscarriage. These findings are possibly explained by inflammation-derived changes in functional properties of the endometrium during the periods of HSV exacerbation or surgical intervention for pregnancy termination.

Thus, at present, knowledge about endometrial microbiota’s significance in causing infertility of unknown origin is being actively developed. Bacteria and/or viruses persisting in the endometrium alter the functional properties of the latter, including its receptivity and embryo implantation [3]. Moreover, it is of high importance to determine the local immune biomarkers in the assessment of certain bacteria and viruses as causative agents of infertility [32], especially in women with recurrent HSV and HPV infection, a history of ectopic pregnancy, and miscarriage. Thus, it is of high relevance to continue further immune biomarker investigation and develop new therapeutic approaches, including not only antibacterial agents or acyclic nucleosides but also immunomodulating drugs.

## 4. Materials and Methods

### 4.1. General Study Design

The study was approved by the ethics committee of “The Research Institute of Obstetrics, Gynecology and Reproductive medicine named after D. O. Ott” (protocol code 108 dated 4 April 2021) and performed at the Department of Assisted Reproductive Technologies. The recruitment period of participants was from November 2019 to May 2022. All participants gave informed written consent for participation. A prospective analysis of the vaginal and uterine microbial composition was carried out. An enzyme-linked immunosorbent assay (ELISA) was performed to verify quantitative levels of α-defensin (DEFa1), the transforming growth factor β1 (TGFβ1), and basic fibroblast growth factor (bFGF2) (Cloud-Clone Corporation (Katy, TX, USA; manufactured in Wuhan, China)) in 65 reproductively-aged women (24 to 40 yo, with a mean age of 34 ± 3.8 years) complaining of secondary infertility in a one-year period. This cohort of patients represented the infertile group. The fertile group of patients included 14 age-matched potential surrogate women, undergoing a comprehensive gynecologic examination at the Department of Assisted Reproductive Technologies. 

Inclusion criteria were the following: age 25–40 years old, confirmed secondary infertility despite regular sexual life, sexual partner’s age less than 50 years old, absence of antibacterial therapy in the past three months, and standard laboratory and clinical evaluation according to the order of the Russian Federation Ministry of Health. Exclusion criteria: age less than 25 years old or more than 40 years old, sexual partner’s age more than 50 years old, two or more sexual partners at the time, tubal, endocrine, male factors of infertility, any gynecologic pathology requiring surgical and/or hormonal treatment, diabetes mellitus or any other endocrine disease requiring hormonal replacement therapy, psychiatric disorders, alcohol and/or drug abuse, history of malignant tumors, HIV-positive patients including those on HAART, and history of viral hepatitis B or C.

### 4.2. Clinical and Social Characteristics of Patients Included

The clinical and social characteristics of patients recruited into the study are presented in Table 5.

Table 6 contains the obstetric and gynecological history of patients in the investigated groups.

### 4.3. Patients Investigation

All patients underwent serological detection of specific antibodies against HIV-1,2, *Treponema pallidum*, viral hepatitis B and C, as well as complete blood count.

During the gynecological examination, vaginal smears were obtained from all patients, separately from the posterior and lateral vaginal fornices. In order not to contaminate the endometrial microbiota with the vaginal one, an Endobrash Standard for Endometrial Cytology (Laboratorie C.C.D., France) was used for endometrial sampling. The latter represents a diamond-shaped brush for the removal of cells from the surface of the endometrium. It retracts completely into the sheath and is closed by a rounded end cap, ensuring the protection of the brush and the sample against contamination.

### 4.4. Sample Testing

For the purpose of DNA extraction, DNA-sorb-AM kits were used (“NextBio” LLC, Moscow, Russia) and DT-96 and DTPRIME amplifiers (“DNA-Technology” LLC, Moscow, Russia) were used for the reaction’s initiation.

A quantitative assessment of the total vaginal and endometrial bacterial mass was carried out using the multiplex Real-Time PCR Detection Kit (“DNA-Technology” LLC, Moscow, Russia). The implemented PCR method is based on the amplification of a target DNA sequence using one biological sample and is expressed in genomic equivalent (GE). GE, in turn, is defined as the amount of DNA necessary to be present in a purified sample to guarantee that all genes will be present. Upon completion of the run, a quantitative analysis of total bacterial mass and genius/species-specific DNA of *Lactobacillus* spp., *Streptococcus* spp., *Streptococcus agalactiae*, *Staphylococcus* spp., *Gardnerella vaginalis*, *Atopobium vaginae*, *Anaerococcus* spp., *Bacteroides* spp./*Porphyromonas* spp./*Prevotella* spp., *Enrerococcus* spp., *Sneathia* spp./*Leptotrichia* spp./*Fusobacterium* spp., *Megasphaera* spp./*Veillonella* spp./*Dialister* spp., *Lachnobacterium* spp./*Clostridium* spp., *Peptostreptococcus* spp., *Bifidobacterium* spp., *Mobiluncus* spp./*Corynebacterium* spp., *Mycoplasma hominis*, *Ureaplasma urealyticum*, *Ureaplasma parvum*, *Candida* spp., *Candida albicans*, *Chlamydia trachomatis*, *Neisseria gonorrhoeae*, *Mycoplasma genitalium*, *Trichomonas vaginalis*, and viruses (human herpes virus type 6 (HHV-6A), human cytomegalovirus (CMV), Epstein–Barr virus (EBV), herpes simplex virus type 1 and type 2 (HSV-1/HSV-2)) was obtained.

Human papilloma virus positivity was determined by the KVANT-21 kit (“DNA-Technology” LLC, Moscow, Russia) based on real-time PCR detection of oncogenic HPV types (16, 18, 26, 31, 33, 35, 39, 45, 51, 52, 53, 56, 58, 59, 66, 68, 73, and 82) and non-oncogenic HPV types (6, 11, and 44). The viral load was expressed in absolute values reflecting the number of HPV DNA copies in the sample, and in relative values indicating the number of HPV DNA copies per human DNA.

### 4.5. Sample Testing

The quantitative measurement of endometrial DEFa1 (SEA124Hu), TGFβ1 (SEA124Hu), and bFGF2 (CEA551Hu) was carried out using a competitive inhibition enzyme immunoassay technique, Cloud-Clone Corporation (Katy, TX, USA; manufactured in Wuhan, China) according to the manufacturer’s instructions. First, the preparation of tissue homogenates was performed. Tissues were rinsed in ice-cold PBS to remove excess blood and minced into small pieces with a glass homogenizer on ice. The resulting suspension was sonicated with an ultrasonic cell disrupter. Then, the homogenates were centrifuged for 15 min at 1000× *g*. The obtained supernates were stored in aliquots at ≤−80 °C. The samples were brought to room temperature before performing the assay. Additionally, 20 μL of 1 M HCI was added to 100 μL of cell culture supernate for TGFβ1 detection. The concentration read of the standard curve was multiplied by the dilution factor—1.4. The optical density was measured at the wavelength of 450 ng/mL.

The standard curves with the log of DEFa1, TGFβ1, and bFGF2 concentrations on the *y*-axis and absorbance on the *x*-axis were created. The minimum detectable dose of TGFβ1 is typically less than 5.7 pg/mL, less than 4.43 pg/mL for bFGF2, and less than 0.125 pg/mL for DEFa1.

### 4.6. Patient Follow-Up

At least 12 months of follow-up counseling or a phone survey of recruited couples was carried out. In June 2022, the median follow-up period of patients was 18.6 months (ranging from 12.5 to 36.2 months).

#### Statistical Analysis

The present study is descriptive, not comparative. All statistical analyses were performed with STATISTICA 10.0 software. The Shapiro–Wilk test was used to evaluate the distribution of the parameters. Normally distributed measurement data were expressed as median (Me) and interquartile range (the difference between the 75th and 25th percentile). Data from the two groups were compared by the Mann–Whitney U test and Wilcoxon rank-sum test. For the prediction of possible outcomes of an uncertain event, the Monte Carlo test was applied. The microbiological concordance to a certain microorganism was estimated by the formula K = [a/a + b)] × 100%, where a is the number of patients testing positive for a specific microorganism both in the vagina and in the endometrium, and b is the number of patients testing positive for a specific microorganism either in the vagina or in the endometrium. Fisher’s exact test (χ^2^) test was used to compare the variables in cases of microorganism combination. For all tests, a *p*-value of <0.05 was considered statistically significant.

## Figures and Tables

**Table 1 ijms-24-07572-t001:** Microbial concordance in the endometrium and in the vagina.

Microorganism	Endometrium	Vagina	Concordance (K, %; a/b)
*Bifidobacterium* spp.	11	15	36.8%; 7/12
*Gardnerella vaginalis*	23	30	33.3%; 13/26
*Mycoplasma hominis*	2	2	33.3%; 1/2
*Lactobacillus* spp.	27	15	28.1%; 9/23
HPV	10	9	26.7%; 4/11
*Enterobacteriaceae* spp.	12	20	23.1%; 6/20
*Bacteroides* spp., *Porphyromonas* spp., *Prevotella* spp.	22	23	21.6%; 8/29
*Enterococcus* spp.	11	16	17.4%; 4/19
*Mobiluncus* spp., *Corynebacterium* spp.	4	9	8.3%; 1/11
*Streptococcus* spp.	7	7	7.7%; 1/12
*Sneathia* spp., *Leptotrichia* spp., *Fusobacterium* spp.	2	13	15.4%; 2/11
*Atopobium vaginae*	10	7	13.3%; 2/13
*Peptostreptococcus* spp.	7	19	13.0%; 3/20
*Megasphaera* spp., *Velionella* spp.	6	14	11.8%; 2/15
*Clostridium* spp., *Lachnobacterium* spp.	3	12	7.1%; 1/13
*Staphylococcus* spp.	0	7	0%; 0/7
*Streptococcus agalactiae*	4	2	0%; 0/6
*Candida* spp.	1	11	0%; 0/12
*Ureaplasma parvum*	2	6	0%; 0/8
*Haemophilus* spp.	0	1	0%; 0/1
*Anaerococcus* spp.	3	3	0%; 0/6

**Table 2 ijms-24-07572-t002:** Endometrial TGFβ1, bFGF2, and DEFa1 concentrations in the investigated groups.

Parameter	Infertile Group	Fertile Group	U	W	Z	*p*
bFGF2, pg/mL	19.931 [16.18; 40.65]	48.8 [40.0; 50.83]	159.5	2304.5	−3.79	<0.0001
TGFβ1, pg/mL	5.28 [2.93; 8.4]	8.02 [6.008; 10.6]	252.0	2397.0	−2.61	0.009
DEFa1, pg/mL	0.746 [0.532; 1.159]	0.615 [0.493; 0.672]	297.5	402.5	−2.022	0.042

U—Mann–Whitney U test, W—Wilcoxon rank-sum test. Data are presented as the median and interquartile range (25th and 75th percentile). bFGF2—basic fibroblast growth factor, TGFβ1—transforming growth factor beta, DEFa1—α-defensin.

**Table 3 ijms-24-07572-t003:** Endometrial TGFβ1, bFGF2, and DEFa1 concentrations among patients with idiopathic infertility in relation to microorganisms verified.

Microorganism/ Positive History	Presence in Endometrium/ Positive History	bFGF2, pg/mL	TGFβ1, pg/mL	DEFa1, pg/mL
Microbiological data				
*Lactobacillus* spp.	Yes (n = 27)	20.3 [16.81; 42.55]	5.28 [2.2; 9.23]	0.83 [0.63; 1.23]
	No (n = 38)	19.71 [17; 25.41]	5.13 [3.22; 7.95]	0.69 [0.52; 0.92]
	stat.	t = 1.0724 *p* = 0.261	t = 1.2 *p* = 0.234	t = 1.818 *p* = 0.068
*Peptostreptococcus* spp.	Yes (n = 7)	40.9 [19.49; 42.6]	5.86 [5.35; 7.21]	0.653 [0.588; 0.687]
	No (n = 58)	19.71 [15.52; 25.79]	4.76 [2.83; 8.5]	0.75 [0.53; 1.12]
	stat.	U = 134.5 *p* = 0.149	t = 0.198 *p* = 0.797	U = 137.5 *p* = 0.171 F = 1562.7 *p* = 0.0011
HPV	Yes (n = 10)	19.34 [12.99; 29.66]	5.57 [4.73; 7.67]	0.97 [0.54; 4.36]
	no (n = 55)	19.93 [17.16; 40.65]	5.12 [2.86; 8.63]	0.75 [0.55; 1.01]
	stat.	U = 232.5 *p* = 0.44	t = 1.75 *p* = 0.086	t = 3.132 *p* = 0.003
Anamnestic data				
Recurrent genital HSV infection	Yes (n = 21)	19.49 [15.52; 42.59]	5.42 [2.05; 9.8]	1.27 [0.9; 5.42]
	No (n = 44)	20.12 [17.32; 38.45]	5.205 [3.22; 8.18]	0.65 [0.49; 0.77]
	stat.	t = 0.573 *p* = 0.57	U = 460.5 *p* = 0.989; F = 9.6 *p* = 0.028	t = 4.255 *p* = 0.008
Ectopic pregnancy	Yes (n = 8)	8.77 [5.4; 19.184]	0.81 [0; 4.21]	0.453 [0.256; 0.73]
	No (n = 57)	20.52 [18.11; 42]	5.28 [3.22; 8.2]	0.75 [0.59; 1.15]
	stat.	U = 80.5 *p* = 0.002	U = 71.5 *p* = 0.0012	U = 93 *p* = 0.005

U—Mann–Whitney U test, F—Wilcoxon rank-sum test, F—Fisher’s exact test. bFGF2—basic fibroblast growth factor, TGFβ1—transforming growth factor beta, DEFa1—α-defensin.

**Table 4 ijms-24-07572-t004:** Composition of endometrial microbiota and biomarker concentrations in patients with pregnancy achieved during the follow-up period.

Microorganism	Patient’s Identification Number							
	6	14	17	23	42	55	56	62
*Lactobacillus* spp.	+				+	+	+	+
*Bifidobacterium* spp.			+			+		
*Gardnerella vaginalis*	+		+	+		+		
*Atopobium vaginae*	+			+		+		
*Bacteroides* spp., *Porphyromonas* spp., *Prevotella* spp.				+		+		
*Enterobacteriaceae* spp.		+	+	+		+		+
*Enterococcus* spp.				+				
*Megasphaera* spp., *Velionella* spp.						+		
*Peptostreptococcus* spp.						+		
*Streptococcus* spp.		+			+	+		+
*Anaerococcus* spp.						+		
Human papillomavirus (HPV)	+							
Immune biomarkers								
bFGF2, pg/mL	3.373	5.397	15.52	25.41	0.675	6.071	18.73	5.399
TGFβ1, pg/mL	49.19	2.052	8.65	3.079	2.345	0	6.3	0
DEFa1, pg/mL	16.12	0.665	2.098	0.996	0	0.25	0.486	0.425
Number of microorganisms	4	2	3	5	2	10	1	3

**Table 5 ijms-24-07572-t005:** Clinical and social parameters in groups investigated.

Parameter	Infertile Group	Fertile Group
Number of patients, n	65	14
Age [min; max], yo	25–37	25–38
Median [25th percentile; 75th percentile], yo	33 [28; 35]	34 [29; 35]
Education:		
Higher education	27 (41.5%)	14 (100%)
Intermediate vocational education	21 (32.3%)	-
Secondary-level education	17 (26.2%)	-
Social status:		
Married	32 (49.2%)	8 (57.1%)
Single	33 (50.8%)	6 (42.9%)
Body mass index (kg/m^2^)		
18.5–19.9	9 (13.8%)	-
20.0–25.0	29 (44.6%)	10 (71.4%)
25.0–29.9	26 (40.0%)	4 (28.6%)
Smoking:		
Yes	22 (33.8%)	3 (21.4%)
No	43 (66.2%)	11 (78.6%)

**Table 6 ijms-24-07572-t006:** Parameters of obstetric and gynecological history in the investigated groups.

Parameter	Infertile Group	Fertile Group
Number of deliveries 1 2	65 (100%) 43 (66.2%) 22 (33.8%)	14 (100%) 0 14 (100%)
History of miscarriage followed by: *uterine abrasion* *medication-induced abortion*	45 (69.2%) 34 (52.3%) 11 (16.9%)	0 0 0
Ectopic pregnancy followed by: *laparoscopic surgery* *laparotomy*	8 (12.3%) 7 (10.8%) 1 (1.5%)	0 0 0
History of IVF failures	8 (12.3%)	0
History of surgical interventions *hysteroscopy* *laparoscopy*	49 (75.4%) 38 (58.5%) 15 (23.1%)	0 0 0
History of viral infections *recurrent* herpes simplex virus *infection* *history of* human papillomavirus	27 (41.5%) 21 (32.3%) 8 (12.3%)	0 0 0

## Data Availability

The data that support the findings of this study are available on request from the corresponding author. The data are not publicly available due to privacy or ethical restrictions.

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
