# Peer review of "Local Immune Biomarker Expression Depending on the Uterine Microbiota in Patients with Idiopathic Infertility"

_ijms, 2023, doi:10.3390/ijms24087572_

Round 1

Reviewer 1 Report

Local Immune Biomarkers Expression Depending on the Uterine Microbiota in Patients with Idiopathic Infertility

Authors outline cohort of 65 female patients with idiopathic infertility, and report patient endometrium samples displayed decreased TGFβ1/bFGF2 and increased DEFa1 compared with fertile patients (14 age matched subjects).  These changes were correlated with presence of Peptostreptococcus spp. and HPV in the uterine cavity.  While uterine infections resulting in female infertility is a well-known phenomenon (PMID: 33091407), this study describes biomarkers that may provide a way for assessing the presence of infection that may be causing infertility.

Questions

Line 90: Fig1 "The rate of patients with various number of microorganisms detected in endometrium and vagina."

Questions: are these patients from both groups combined into a single figure?  Could you provide a figure with the two patient groups separated?

Line 101: Table 1.

Question: I'm not sure what you are doing statistics here for?  It seems like you are comparing different microorganisms between the groups?  Text is not clear?

Line 122: Table 4

Comment: Please state in table legend how data is presented?  Some data looks like Median and interquartile range, while other data looks like Mean ± SD?  Please choose one way to present your data and stick with it.  Covert all data into same format please.

Line 133: Table 5

Comment: would like to see the TGFβ1, bFGF2 and DEFa1 data for each patient also included in this table.

Line 245:  Main group vs Comparison group

Suggestion:  may be easier for reader to change "Main Group" to "Infertile Group" and Comparison group to "Fertile Group"

Line 248: Table 7. Would be nice to see this table include both "main group" and "comparison group"

Line 229: "The comparison group of patients included 14 age-matched potential surrogate women"

Question: where these women followed up to confirm their fertility?

Suggestions

Line 29: "At present, there is evidence indicating the presence of endometrial microbiota."

Citation(s) needed here.

Line 46: "Defensins represent cationic peptides consisting of 18-45 amino acids, active against bacteria, fungi and a large number of both enveloped and non-enveloped viruses."

Citation(s) needed here.

Line 49: "More recently, the antiviral activity of some defensins was demonstrated."

Citation(s) needed here.

Line 55: "participates in the development and homeostasis of most human body tissues."

Citation(s) needed here.

Line 73: Suggest change "no microorganisms only in 6 (9.23%) endometrial samples" to "the lack of microorganisms in 6 (9.23%) endometrial samples"

Line 74: Suggest change "not identified at all." to "not found in any endometrial samples.

Line 75: Suggest change "Lactobacilli solely were found in 6 (9.23%) samples" to "Lactobacilli alone was found in 6 (9.23%) samples"

Line 76: Suggest change "determined only in 1" to "found in only 1"

Line 90: Fig1

Please Increase size of legend boxes.  It's hard to see colours in the small legend boxes

Line 114: Table 3

Please state in table legend data is presented as Median and interquartile range [25th and 75th percentile]

Line 151: "In our study, only 9.2% of the endometrium samples did not contain microorganisms"

Need to be clear, this was for the "Main Group".  In the normal group 64% did not contain microorganisms.

Author Response

Dear Reviewer, thank you so much for high appreciation of our research and your comments.

Questions

Q1. Line 90: Fig1 "The rate of patients with various number of microorganisms detected in endometrium and vagina."

Questions: are these patients from both groups combined into a single figure?  Could you provide a figure with the two patient groups separated?

A1. Figures were added.

Q2. Line 101: Table 1.

Question: I'm not sure what you are doing statistics here for?  It seems like you are comparing different microorganisms between the groups?  Text is not clear?

A2: After careful consideration, authors agreed to remove the table from the manuscript.  

Q3. Line 122: Table 4

Comment: Please state in table legend how data is presented?  Some data looks like Median and interquartile range, while other data looks like Mean ± SD?  Please choose one way to present your data and stick with it.  Covert all data into same format please.

A3. Corrected

Q4. Line 133: Table 5

Comment: would like to see the TGFβ1, bFGF2 and DEFa1 data for each patient also included in this table.

A4. Information was added

Q5. Line 245:  Main group vs Comparison group

Suggestion:  may be easier for reader to change "Main Group" to "Infertile Group" and Comparison group to "Fertile Group"

A5. Changes have been made

Q6. Line 248: Table 7. Would be nice to see this table include both "main group" and "comparison group"

A6: Information was added

Q7. Line 229: "The comparison group of patients included 14 age-matched potential surrogate women"

Question: where these women followed up to confirm their fertility?

A7. Information was added.

Suggestions

Line 29: "At present, there is evidence indicating the presence of endometrial microbiota."

Citation(s) needed here.

A: Citation was added.

Line 46: "Defensins represent cationic peptides consisting of 18-45 amino acids, active against bacteria, fungi and a large number of both enveloped and non-enveloped viruses."

Citation(s) needed here.

A: Citation was added.

Line 49: "More recently, the antiviral activity of some defensins was demonstrated."

Citation(s) needed here.

A: Citation was added.

Line 55: "participates in the development and homeostasis of most human body tissues."

Citation(s) needed here.

A: Citation was added.

Line 73: Suggest change "no microorganisms only in 6 (9.23%) endometrial samples" to "the lack of microorganisms in 6 (9.23%) endometrial samples"

A: Corrected

Line 74: Suggest change "not identified at all." to "not found in any endometrial samples.

A: Corrected

Line 75: Suggest change "Lactobacilli solely were found in 6 (9.23%) samples" to "Lactobacilli alone was found in 6 (9.23%) samples"

A: Corrected

Line 76: Suggest change "determined only in 1" to "found in only 1"

A: Corrected

Line 90: Fig1

Please Increase size of legend boxes.  It's hard to see colours in the small legend boxes

A: Corrected

Line 114: Table 3

Please state in table legend data is presented as Median and interquartile range [25th and 75th percentile]

A: Corrected

Line 151: "In our study, only 9.2% of the endometrium samples did not contain microorganisms"

Need to be clear, this was for the "Main Group".  In the normal group 64% did not contain microorganisms.

A: Corrected

Reviewer 2 Report

1. The review includes enough cases in the first lot, but the control group includes a few cases, all with a high educational level.

2. More detail on the methods used in the study, which would help readers understand the strengths and limitations of the research findings.

3. Microbiota also changes depending on the environment, so the control lot is not balanced. It is not specified how the uterine microbiota was collected, in order not to contaminate the sample with the vaginal one.

4. As for the inclusion criteria, it is not specified whether the patients have had antibiotic treatments and for how long.

5. It would be beneficial to include more recent studies and research on the topic.

6. More background information and clearer discussions of the significance of the findings. 

Author Response

Dear Reviewer, thank you so much for high appreciation of our research and your comments.

  1. The review includes enough cases in the first lot, but the control group includes a few cases, all with a high educational level.

A: Indeed, the control group is smaller, mainly because of our focus on women complaining of infertility, whereas controls were fertile. After consideration, we decided to change the names of the groups for the readers’ convenience.

  1. More detail on the methods used in the study, which would help readers understand the strengths and limitations of the research findings.

A: Information was added

  1. Microbiota also changes depending on the environment, so the control lot is not balanced. It is not specified how the uterine microbiota was collected, in order not to contaminate the sample with the vaginal one.

A: This information can be found in section 4.3. Patients investigation

  1. As for the inclusion criteria, it is not specified whether the patients have had antibiotic treatments and for how long.

A: Information was added.

  1. It would be beneficial to include more recent studies and research on the topic.

A: Information was added.

  1. More background information and clearer discussions of the significance of the findings. 

A: Information was added.

Reviewer 3 Report

This is a descriptive study where it is described the qualitative and quantitative composition of the resident endometrial and vaginal microbiota, in reproductively-aged women and age-matched potential surrogate women. Also, quantitative measurement of DEFa1, TGFβ1, and bFGF2 was carried out in the endometrium. Despite not being a deep comparative study, and a larger number of cases being desirable, this is a necessary and useful study.

Some minor changes are encouraged: 

- Line 298 needs a reference.

- The indication of the number of microorganisms found needs a correction: indicate with words when there are 1 to 10 microorganisms (for example, four microorganisms instead of 4 microorganisms), and preceding "microorganisms" you may need a descriptive word (different, genius, species...).

Author Response

Dear Reviewer, thank you so much for high appreciation of our research and your comments.

Q1. Line 298 needs a reference.

A1. This particular line contains inclusion criteria. Please, let me know which particular line needs a reference?

Q2. The indication of the number of microorganisms found needs a correction: indicate with words when there are 1 to 10 microorganisms (for example, four microorganisms instead of 4 microorganisms), and preceding "microorganisms" you may need a descriptive word (different, genius, species...).

A2: Corrected.